# Customized Millimeter Wave Channel Model for Enhancement of Next-Generation UAV-Aided Internet of Things Networks

**DOI:** 10.3390/s24051528

**Published:** 2024-02-27

**Authors:** Faisal Altheeb, Ibrahim Elshafiey, Majid Altamimi, Abdel-Fattah A. Sheta

**Affiliations:** Electrical Engineering Department, King Saud University, Riyadh 12372, Saudi Arabia; ishafiey@ksu.edu.sa (I.E.); mtamimi@ksu.edu.sa (M.A.); asheta@ksu.edu.sa (A.-F.A.S.)

**Keywords:** access point (AP), internet of things (IoT), sensor node (SN), millimeter-wave channel modeling, 3D networks, UAV-enabled networks, clustered delay line (CDL)

## Abstract

The success of next-generation Internet of Things (IoT) applications could be boosted with state-of-the-art communication technologies, including the operation of millimeter-wave (mmWave) bands and the implementation of three-dimensional (3D) networks. With some access points (APs) mounted on unmanned aerial vehicles (UAVs), the probability of line-of-sight (LoS) connectivity to IoT nodes could be augmented to address the high path loss at mmWave bands. Nevertheless, system optimization is essential to maintaining reliable communication in 3D IoT networks, particularly in dense urban areas with elevated buildings. This research adopts the implementation of a geometry-based stochastic channel model. The model customizes the standard clustered delay line (CDL) channel profile based on the environmental geometry of the site to obtain realistic performance and optimize system design. Simulation validation is conducted based on the actual maps of highly dense urban areas to demonstrate that the proposed approach is comprehensive. The results reveal that the use of standard channel models in the analysis introduces errors in the channel quality indicator (CQI) that can exceed 50% due to the effect of the environmental geometry on the channel profile. The results also quantify accuracy improvements in the wireless channel and network performance in terms of the CQI and downlink (DL) throughput.

## 1. Introduction

The Internet of Things (IoT) provides the basis for developing future smart urban [1] and rural areas [2]. The design of IoT networks should maintain the simplicity of the nodes while satisfying the function requirements in terms of sensing, computation, and communication tasks. Depending on the application, the design should consider various constraints related to energy requirements, latency, and computational resources.

IoT networks would be enabled by next-generation systems, including 6G, which is promising to enable use cases that have not been possible due to sophisticated technical requirements [3,4,5]. Most existing commercial networks rely on sub-6GHz bands [6]. Such bands are not sufficient to meet future data rate requirements due to scarcity; they have been extensively used in existing technologies [7], and they require antenna size that can increase the node size, particularly when a large number of array elements is desired to enhance beamforming capability. Next-generation networks will utilize millimeter-wave (mmWave) frequencies, which are abundant and can deliver ultra-high throughput. Experimental trials have demonstrated that mmWave can perform long-range communications in line-of-sight (LoS) scenarios [8]. The mmWave is expected to play a vital role and become essential in 6G, along with new spectrum bands such as terahertz (THz) and free space optics (FSO). Despite its potential, mmWave suffers from high path and penetration losses in non-line-of-sight (NLoS) scenarios, such as in an urban area with many high-rise buildings.

To alleviate the propagation challenges of mmWave bands, a three-dimensional (3D) network architecture could be adopted by installing access points (APs) on unmanned aerial vehicles (UAVs) that can be easily controlled and relocated to achieve LoS connectivity and enhance coverage and throughput [9]. Such an approach coincides with the vision of 6G, which is expected to integrate aerial elements, such as UAVs, with terrestrial networks to provide high-quality and cost-efficient mobile services [10].

UAVs coincide with the cellular concept since low altitude combines coverage superiority and confined cell radius [11,12]. The coverage area can be controlled by altitude adjustment; however, an enlarged coverage area will result in high path loss. Therefore, the optimum placement of UAVs depends on the design objective. UAVs can either support mobility or hover above the coverage area. The mobility of UAVs creates a Doppler shift, which causes severe inter-carrier interference at higher transmission frequencies [13]. Trajectory planning and control must be investigated to avoid interference in such a scenario. Regulations on the operation of UAVs in urban areas can impose restrictions in terms of the allowable areas and altitudes [14].

Despite the conceptual simplicity of using UAVs to enhance IoT networks, the actual implementation requires advanced measures to control system parameters and enable the functionality of such networks. Artificial intelligence (AI) techniques have been shown to play a major role in this regard [15]. Advanced optimization techniques, such as genetic algorithms, have been suggested to enhance energy efficiency by controlling the UAV trajectory [16]. The role of UAVs in IoTs exceeds the communication part to include a computational enhancement by enabling mobile edge computing to offload tasks from IoT devices and optimize the management of energy and time resources [17,18]. Non-orthogonal multiple access (NOMA) techniques, along with UAV networks, have been proposed for ultra-reliable low-latency communication (URLLC) applications [9]. UAVs have also been suggested to play new roles in IoT networks, such as wireless powering of IoT devices [19].

The success of adopting UAV-enabled networks operating in mmWave bands depends on the accurate characterization of channel behavior. For example, the LoS blockage probability of UAV links in urban deployment is presented in [20]. Effective data acquisition for wireless sensors with obstacles is presented in [21]. A detailed overview of channel model literature is shown next, followed by an illustration of the objectives and contributions of this paper.

### 1.1. Channel Models

Path loss channels are considered in most studies to model an air-to-ground (ATG) wireless link, along with a network architecture that depends solely on aerial wireless access elements, such as UAVs. In [22], the deployment of an aerial cloud radio access network (cloud-RAN) is considered to minimize the energy consumption of UAVs. The location, coverage radius, and functional split option of each UAV are jointly optimized, and a path loss channel model is considered. A cloud-RAN in which devices or user terminals (UTs) on the ground are served by UAVs to offload their tasks is considered in [23] with a path loss channel. The aim is to maximize the sum rate of all the UTs by jointly optimizing the UT association, transmit power, and UAV placement. In [24], mobile flying base stations (BSs) mounted on UAVs with a path loss channel are exploited to maximize data throughput by controlling the travel directions of UAVs. Whereas standard stochastic channel models have been widely used to study terrestrial networks, some studies have extended this model to ATG cases, as in [25,26,27], to study coverage probability rates when UAVs are used for the purpose of providing cellular connectivity. The advantage of the stochastic channel model is its ability to capture spatial randomness and provide an analytical model that applies, on average, to all cellular networks’ realizations [28]. In [29], the coverage probability of UAV-assisted cellular networks is evaluated in urban and rural areas using a stochastic geometry-based model. Study [30] focuses on providing cellular connectivity to offshore areas for search and rescue operations. It considers a simple two-ray channel model to compare the performance of UAVs acting as relays or BSs in terms of data rate and latency.

Hybrid network architectures typically use UAVs to complement terrestrial BSs. The authors in [31] considered a path loss channel in studying the use of UAVs to complement terrestrial cloud-RAN in scenarios where a subset of terrestrial BSs breakdown. Interestingly, it is found that using UAVs to compensate for the failed BSs can deteriorate performance in some cases, as compared to using only the remaining working BSs. This is because the presence of UAVs can increase the interference level. With a path loss channel, the feasibility and performance of a multi-tier drone architecture are investigated in [13] in terms of spectral efficiency, optimal drone proportion, and altitude. Load conditions in which the deployment of UAVs can be beneficial are identified. The authors in [32] proposed a drone-cell deployment framework, considering a path loss channel, to alleviate overload conditions caused by flash crowd traffic and boost the capacity of the ground network. The interference caused by proximate drone cells and ground small cells is not considered. A clustering method is employed to identify the required number of drone cells and their locations. Table 1 summarizes existing frameworks in UAV-assisted communication systems.

Several channel models have been developed by standards organizations, academia, and research groups to characterize a wireless medium at mmWave and sub-6 GHz bands in different propagation environments. Table 2 summarizes these channel models according to [33].

The 3GPP specified 5G channel models for frequencies from 0.5 to 100 GHz in various scenarios, such as urban micro (UMi) and urban macro (UMa), with outdoor-to-outdoor (O2O) and outdoor-to-indoor (O2I). Two stochastic channel models have been defined for 5G link-level evaluations: the clustered delay line (CDL) and the tapped delay line (TDL). Both models have predefined profiles called A, B, C, D, and E. Profiles A, B, and C are for NLoS scenarios, whereas D and E represent LoS channels. Standard channel profiles can be found in [34]. Similar to the traditional 4G or 3G channels, TDL models consider taps with Rayleigh fading characteristics, where delay and power profiles are assigned to each tap. On the other hand, CDL models account for clusters of rays at the transmitter and receiver sides. In addition to delay and power level values, angles are assigned to each cluster. Thus, CDL models are suitable for 3D propagation simulations. The METIS identified 5G requirements for a shorter frequency range, from 2 to 60 GHz, focusing on outdoor square and indoor shopping mall scenarios. The QquaDRiGa supports a frequency range from 0.45 to 100 GHz. However, it is currently well aligned with the TR 36.873 study on the 3D channel model for long-term evolution (LTE) and will be extended to include 5G requirements. The NYUSIM is built on radio propagation measurements for a frequency range from 0.5 to 150 GHz, covering a variety of scenarios as the 3GPP [33].

### 1.2. Motivation

Despite the promising results obtained in the studies summarized in Table 1, most studies have not considered geometry-based channel models. Distance-dependent path loss channel models are based on media such as free space, rain, fog, or gas. They are not realistic because ATG channels depend on the elevation angle as well [38]. The elevation angle plays a more important role in high-frequency communications, such as mmWave, because the antenna beamwidth is narrow. Standard stochastic channel models address multipath fading and typically do not capture the spatial variation in coverage between different environments, such as urban or rural [29]. These models are generic and not intended for a specific realization. Neither the path loss nor the standard stochastic channel model require prior knowledge of environmental geometry. Because of the geometry, a UT can receive LoS, strong NLoS, and multiple reflected components that cause multipath fading [32]. Standard stochastic channel models used for sub-6 GHz assume two-dimensional (2D) propagation; such an assumption is not valid for mmWave [39]. Therefore, 3D geometry-based stochastic channel models are essential for next-generation IoT systems and smart cities, which will be enabled by the mmWave and 3D networks. The use of such channels in UAV-based IoT networks, such as in smart cities and highly populated urban areas, needs to be sought.

Geometry-based spatial channel models, whether deterministic or stochastic, typically define scatterers that reflect signals transmitted to a receiver. They are developed to better represent beamforming and multiple-input-multiple-output (MIMO) links since other channel models do not account for array geometries and array responses. Among the spatial channel models is the ray tracing model, in which the locations of scatterers are specified using building location information. In spatial models, waves are considered simple particles; hence, reflection and scattering are approximated using a simple geometry. Because the geometry of the environment is known in practice, spatial channel models can provide more accurate results as compared with general channel models. While a simple two-ray channel model is considered in [30], such a model is specific and limited to half-space regions, such as over-the-sea communications. While there are several channel models illustrated in Table 2 for 5G and beyond, the 3GPP is widely adopted [33]. It identifies 5G requirements, supports a relatively wide frequency range, and covers a variety of scenarios.

### 1.3. Objective, Contribution, and Structure

This study aims to investigate a geometry-based stochastic channel-modeling scheme for next-generation 3D IoT networks in terms of wireless channel quality and network performance. The 3D IoT networks operate at mmWave and are complemented by aerial APs mounted on UAVs. This research contributes to next-generation mobile networks in several aspects, as follows:Unlike existing studies that use path loss or standard stochastic channel models, a sophisticated geometry-based stochastic channel modeling scheme is developed.Using a spatial channel model, standard CDL channel profiles defined by 3GPP are customized based on the geometry of the environment. To the best of our knowledge, this is the first study that incorporates a spatial channel model into a CDL channel in the context of UAV-assisted networks.Although many studies consider a homogeneous setup in which BSs are located in hexagonal grids and UTs are uniformly distributed, this research follows a more realistic geometry-dependent approach to modeling the locations of APs and sensor nodes (SNs).The evaluation of cellular network performance can be computationally complex, particularly for heterogeneous networks with many UTs. This study proposes a simplified approach by identifying the relationship between radio conditions and the modulation and coding scheme (MCS). Specifically, the signal-to-interference-plus-noise ratio (SINR) and channel profiles are mapped to channel quality indicator (CQI) indices, and such a mapping approach can be generalized to other environments.

In our previous study [40], a CQI versus SINR table was proposed for a single UT. This work is extended in four main directions. First, the areas of interest are characterized by statistical parameters by considering many UTs, or equivalently, SNs. Second, the altitude of the UAVs is optimized to ensure LoS and achieve the highest spectral efficiency. Third, the network performance is evaluated in terms of downlink (DL) throughput. Fourth, it is shown that results accuracy can be improved considerably when customizing CDL channel profiles instead of following the standard channels.

The remainder of this paper is organized as follows. In Section 2, a reference model consisting of terrestrial APs and state-of-the-art technology is presented. Subsequently, a new system model is proposed to incorporate flying APs in the form of UAVs. Section 3 presents the simulation results, and Section 4 provides concluding remarks.

## 2. System Model

The evolution from 5G to 6G could be gradual, and both technologies may coexist in the future for a while. Therefore, it can be assumed that the early phase of 6G standards will offer the possibility of adapting the current deployments of 5G and sharing similar spectrum bands. In this study, the most recent 5G specifications are followed, and the term UAV is used to refer to an AP mounted on a UAV unless otherwise specified. The reference system consisted of terrestrial APs. It is modeled according to [34], and an O2O 3D UMi scenario is considered.

### 2.1. Areas of Interest

Two different urban areas are considered in the simulation. The first area (Area-A) is the city center of Hong Kong, and the second area (Area-B) is Manhattan, New York. In addition to being realistic examples of urban areas, both areas are characterized by a large number of buildings with various towering heights, which makes them appropriate choices for developing multipath channel models. Maps are obtained from the OpenStreetMap (OSM) project [41], which includes areas’ features, such as street and building heights. MATLAB SiteViewer 2023 is used to import and visualize 3D building data.

### 2.2. Channel Model

Geometric angles used for radio channel modeling include the zenith angle *θ* and the azimuth angle ϕ in a Cartesian coordinate system, where *θ* = 0° points to the zenith and *θ* = 90° points to the horizon.

First, a ray tracing channel model is used to compute the parameters of each ray, including path loss, delay, and angles. To identify the properties of each ray, the shooting and bouncing rays (SBR) method, which supports up to eight reflections, is considered. The SBR method is widely used in practical propagation modeling [42]. To presume a realistic model, the effect of materials is considered by assuming that all buildings are made of bricks, with permittivity and conductivity as in [43]. In addition, weather impairments are added to the propagation model.

While ray tracing finds individual rays between the AP and SN, the CDL channel models ray clusters. Therefore, the information retrieved for each ray configures the cluster average values of the CDL channel. The path losses obtained from ray tracing are considered the average path gains of the CDL. The elevation angles returned by ray tracing are converted to zenith angles that are used by the CDL, as follows:(1)θ=90°−γ,
where θ is the zenith angle (in degrees), and *γ* is the elevation angle (in degrees) at either the transmitter or receiver. Finally, it is worth noting that CDL introduces small-scale fading because it models SN movement as compared with ray tracing, which is used for static analysis.

To model multiple interactions with the scattering media in the downlink, departure angles are considered for the first scatterers interacted with from the transmitting side, and arrival angles are for the last bounce scatterers. For an uplink, the arrival and departure parameters can be swapped. In addition to the geometry parameters, small-scale parameters, such as cluster delays and power, need to be considered. Cluster delays are normalized by subtracting the minimum delay and sorted in ascending order, as follows [34]:(2)$$\tau_{n}=\mathrm{sort}\left(\tau_{n}^{\prime }-\min\left(\tau_{n}^{\prime }\right)\right)$$
where $\tau_{n}^{\prime }$ is the absolute delay (in seconds) for cluster *n*. For a LoS condition, additional scaling of delays is made as follows [34]:(3)$$\tau_{n}^{LOS}=\frac{\tau_{n}}{0.7705-0.0433K+0.0002K^{2}+0.000017K^{3}},$$
where $\tau_{n}$ is the normalized delay, and *K* is the Ricean *K*-factor (in dB). Cluster powers are normalized such that the sum of all cluster powers is equal to one, as in [34]:(4)$$P_{n}=\frac{P_{n}^{\prime }}{\sum_{n=1}^{N}P_{n}^{\prime }},$$
where $P_{n}^{\prime }$ is the absolute power in the linear scale for cluster n and N is the number of clusters. For a LoS condition, an additional specular component is added to the first cluster, and the cluster powers are normalized as follows:(5)$$P_{n}=\frac{1}{K_{R}+1}\frac{P_{n}^{\prime }}{\sum_{n=1}^{N}P_{n}^{\prime }}+\delta \left(n-1\right)\frac{K_{R}}{K_{R}+1},$$
where $\delta \left(.\right)$ is Dirac’s delta function and $K_{R}$ is the Ricean *K*-factor in linear scale. Therefore, for LoS cases, the CDL model splits the first path into two components: one being LoS and the other having a Rayleigh fading characteristic. This results in a combined Ricean fading characteristic.

Instead of using the generic profiles available in the standards, such an approach can provide a customized CDL channel profile that represents the intended environment by considering its geometry.

### 2.3. Terrestrial Access Points (APs) and Sensor Nodes (SNs)

The environment topology influences the wireless channel since it can impose restrictions on the locations where the APs may be installed and the locations where SNs may exist. APs and SNs are modeled following the traditional models of BSs and UTs in modern cellular systems. Leveraging the work in [44], APs and SNs are randomly distributed in outdoor areas according to a homogeneous Poisson point process (PPP). This allows studying the heterogeneity relation with AP or SN density. While modeling AP locations as PPP can result in some APs being very close to each other, which may not reflect real AP locations, this model can provide similar performance results when compared with real cases [45,46]. To obtain a realistic distribution, an AP that is less than 200 m apart from an adjacent AP is eliminated. Frequency reuse-3 is considered in the system. AP sites are modeled such that each consists of three sectors that are 120° apart, and an antenna is assigned to each sector. Because indoor users and sensors can offload their data traffic through in-building solutions and Wi-Fi, they can be neglected for the purpose of this study. The height of each AP and each SN is chosen to be 10 m and 1.5 m, respectively, according to [34]. The MATLAB Phased Array System Toolbox 2023 is used to implement the antenna element pattern defined in [47]. To increase the directional gain and SINR, the AP antenna element is used to form an 8-by-8 rectangular antenna array. On the other hand, a 2-by-2 rectangular antenna array is considered for each SN.

### 2.4. UAV-Mounted Access Points (UAVs)

The proposed system comprises terrestrial and flying APs. The use of UAVs is considered to mount APs at desirable altitudes, thus increasing the probability of LoS with SN on the ground. It is assumed that 50% of the existing terrestrial APs can be replaced by UAVs. The selection of such APs can vary depending on the design objective. For this research, underperforming APs with low SNs associations are replaced by UAVs to alleviate the load on congested APs while enhancing wireless channels by increasing the number of SNs with LoS connectivity. Due to size and payload limitations, each UAV cell is equipped with a 2-by-2 transmitter, as compared to 8-by-8 in terrestrial AP. A two-tier system is considered for simplicity; terrestrial APs have a common height of 10 m, while the UAVs hover at a specified altitude. Although the selection of a terrestrial AP height is based on the 3GPP UMi evaluation scenario, the altitude of UAVs needs further study. The UAV altitude is optimized to ensure LoS and achieve the highest spectral efficiency.

### 2.5. Cellular System

A total of 200 MHz in the band n257 (28 GHz) is assumed to be available for use. This band is considered since it has industry support for the deployment of mmWave 5G networks. The channel bandwidth is chosen to be 50 MHz, with the shortest subcarrier spacings (SCS) applicable in the frequency range 2 (FR2), which is 60 KHz.

SN cell association is based on maximum receive signal strength (RSS). The SN receiver considers the cell antenna from which the largest signal power is received as the signal source. The remaining cell antennas act as sources of interference.

If round-robin scheduling is considered, users, or equivalently sensors, in the same cell are served in a sequence, and an equal bandwidth allocation policy among users in the same cell is assumed. Following the work in [45], the user spectral efficiency (USE) under full-load conditions can be obtained based on the Shannon capacity formula as follows:(6)$${USE}_{j}=\frac{1}{N_{i}}{log}_{2}\left(1+{SINR}_{j}\right)$$
where $N_{i}$ is the number of active users in cell *i*, and ${SINR}_{j}$ represents the signal-to-interference-plus-noise ratio for user j. Cell spectral efficiency (CSE) for cell i can be obtained by calculating the sum of users’ spectral efficiencies in that cell, as follows:(7)$${CSE}_{i}=\sum_{j\in i}{USE}_{j}$$

The area spectral efficiency (*ASE*) can be calculated as the weighted arithmetic mean of the *CSE* for all cells as follows:(8)$$ASE=\sum {\frac{N_{i}}{N}CSE}_{i}$$
where N is the number of users in the area.

## 3. Simulation and Results

This section begins with the procedure followed to perform the simulation. Various simulation parameters are listed. Subsequently, for illustration purposes, a simplified example that considers only two APs and two SNs is demonstrated. Finally, the simulation results for multi-sensor realistic environments, namely area-A and area-B, are presented and discussed.

### 3.1. Procedure

The following steps illustrate the procedure for our simulation.

A confined urban area is selected as the simulation environment. Subsequently, the characteristics of the area, such as boundary coordinates, streets, and building locations and heights, are obtained from the OSM. For this research, two different urban areas, named area-A and area-B are selected.Within the simulation environment, APs and SNs are randomly distributed in open areas according to a homogeneous PPP with specific densities. Coordinates are assigned to each AP and SN.The Phased Array System Toolbox 2023 in MATLAB is used to implement the antenna arrays for APs and SNs.MATLAB SiteViewer 2023 is used to perform various operations in 3D, as follows:a.Import and visualize the simulation environment, APs transmitters, and SNs receivers in 3D.b.Compute the distances between nodes, such as APs and SNs.c.Perform ray tracing analysis by computing and plotting propagation paths between APs and SNs. The properties of each ray are computed, including the power loss, delay, azimuth angle of departure (AoD), azimuth angle of arrival (AoA), connection type (LoS or NLoS), and number of reflections in the case of NLoS.d.Compute RSS and SINR at each SN.The number of SNs attached to each cell is computed considering the maximum RSS cell association. The USE, CSE, and ASE are then calculated according to (6)–(8).Using the output of the ray tracing analysis, customized CDL channel profiles are constructed for each channel.For each channel profile, a model is developed for measuring the throughput of the 5G physical downlink shared channel (PDSCH) using MATLAB [48]. The model is customized to compute the maximum CQI and MCS that result in a block error rate (BLER) not exceeding 0.1. It is worth noting that the BLER may exceed 0.1 if a channel profile has poor radio conditions, such as low SINR, even if the lowest CQI and MCS are selected.After identifying the channel profiles and their CQI indices, the results are used in a model developed in MATLAB [49] to compute the network performance in terms of DL throughput considering the round-robin scheduling scheme. In this step, all the results of the reference model are obtained.The output of step 5 is used to identify the APs with the lowest SN association in order to replace them with UAVs, with the assumption that 50% of existing APs can be replaced by UAVs.Starting from an altitude of 15 m for the UAVs, steps 3, 4, and 5 are repeated to identify the optimum altitude that ensures LoS and results in the maximum ASE.After identifying the optimal altitude, steps 6, 7, and 8 are executed to obtain the results for the UAV-assisted model.

### 3.2. Parameters

Table 3 lists the various parameters used for the simulation, following the guidelines in [47] for evaluating 5G radio technologies in dense urban environments.

### 3.3. Illustration

An auxiliary example is presented for illustration purposes. For simplicity, this example considers two APs and two SNs; SN-2 can have a LoS connection with AP-1, and SN-1 may not have a LoS connection with any AP. Cell association based on the maximum RSS shows that SN-1 is served by AP-2, and SN-2 is served by AP-1. Table 4 summarizes the radio conditions for each SN. It should be noted that USE and CSE are equal because each cell serves a single SN. The properties of each ray received by SN-1 are then processed to form a CDL channel profile, called F, for NLoS.

The parameters of profile F can be found in Table 5, where ZoD and ZoA are the zenith of departure and zenith of arrival, respectively. Other parameters, namely c_ASD_, c_ASA_, c_ZSD_, and c_ZSA_, represent the cluster angles spread (in degrees). XPR represents the cluster cross-polarization power ratio (in dB). Following parameter values in [34], cluster angles spread and XPR are chosen as per Table 7.5-6, and cluster delays are scaled to obtain a desired delay spread (DS) of 66 ns as per Table 7.7.3-2. Another profile, called G, is defined for SN-2, which receives a LoS connection following a similar procedure. Cluster delays are scaled to obtain DS of 32 ns as per Table 7.7.3-2 in [34]. The parameters of profile G are presented in Table 6. Notably, the CDL model splits the first path into two components for LoS cases. Moreover, it is expected that the LoS ray will dominate the effects of other paths. Clusters with less than 25 dB power compared to the maximum cluster power can be removed, and the scaling factors do not need to be changed after cluster elimination [34]. After defining the CDL channel profiles F and G, the two profiles are used to compute their CQIs and the corresponding MCSs for 0.1 maximum BLER, considering Table 5.2.2.1-3 in [50].

Table 7 lists the CQI and MCS for channel profiles F and G. Finally, the network performance in terms of DL throughput is computed. Since each cell serves a single SN, scheduling has no effect in this case. Whereas the DL throughput for UT-1 is 1.12 megabits per second (Mbps), SN-2 achieves a DL throughput of 26.28 Mbps. This example shows the superiority of the LoS channel in terms of RSS, SINR, CQI, and DL throughput.

### 3.4. Area-A

Following a similar approach, the simulation is performed on the entire system of the first area of interest, which is the city center of Hong Kong. The environmental area is chosen to be 1 km^2^, with 54 cells located at 18 different locations and 255 SNs. It is found that 137 of the channels are NLoS, 86 are LoS, and 2 SNs are not connected. A CDL channel profile is defined for each SN. Figure 1 illustrates the number of sensors attached to each AP site, and it can be seen that AP 1, 3, 7, 17, and 18 collectively serve 106 SNs, approximately 42% of the sensors. This is because these APs are located in areas where sensor density is high. Underperforming APs with a low SNs association are highlighted in red. The ASE is typically reported in bps/Hz/km^2^; however, it is not necessary to account for the area because the simulation is performed over an area of 1 km^2^. Equation (8) is used to calculate the ASE, which is found to be 3.417 bps/Hz.

Underperforming APs with low SNs associations, highlighted in red in Figure 1, are chosen to be replaced by UAVs. Figure 2 shows the relationship between ASE and UAV altitude in our simulation environment. It can be seen that an altitude of 50 m for the UAVs results in a maximum ASE of 6.973 bps/Hz, an improvement of 104% in the ASE compared to using only terrestrial APs. In terms of association, 140 SNs are served by UAVs, and 85 SNs are connected to terrestrial APs. It is also found that one of the remaining terrestrial APs (one site with three cells) does not serve any SN and can be relocated to serve another area.

Since the simulation is performed over many SNs, statistical distributions are considered to fit and model different results. Whereas SN distributions in terms of RSS and SINR are modeled according to the normal (Gaussian) distribution, distributions based on CQI and DL throughput are presented in a cumulative ascending manner. The normal (Gaussian) fit shows that the mean of RSS is −94.32 dBm with a standard deviation of 28 dBm for the case of using only terrestrial APs. On the other hand, the mean is found to be −63.78 dBm and the standard deviation is 12 dBm for the case of using UAVs with terrestrial APs. The mean of the SINR fit is found to be −0.55 dB, with a standard deviation of 26 dB for the case of using only terrestrial APs. For the case of using UAVs with terrestrial APs, the mean is improved to 20.69 dB with a standard deviation of 11.6 dB. Interestingly, it is found that the absolute means of RSS and SINR are equal to the corresponding means of their normal (Gaussian) fits, with the percentage of error not exceeding 1%.

Figure 3 shows the number of SNs reporting each CQI index. When relying solely on terrestrial APs, more than 80 SNs reported the lowest CQI of 1. This means that they are operating at the lowest MCS; meanwhile, they cannot ensure that the BLER does not exceed 0.1. On the other hand, only eight SNs report a CQI index of 1 when the UAVs are considered in conjunction with terrestrial APs.

Finally, the performance is evaluated in terms of DL throughput, considering slot-based round-robin scheduling. The distribution of the SNs based on the DL throughput is shown in Figure 4. SNs with relatively low DL throughput may either have poor radio conditions, such as a low SINR, or they may share a common cell with many other SNs. Table 8 summarizes the results and shows the improvement achieved when UAVs are used in conjunction with terrestrial APs. The averages of the CQI and network DL throughput per SN are listed in the table.

As can be observed from simulation results, the partial replacement of terrestrial APs by UAVs can enhance the overall system performance, despite the fact that UAVs have smaller antenna arrays than terrestrial APs.

#### 3.4.1. Customized versus Standard Channel Profiles

The effect of constructing customized CDL channel profiles is investigated. A simulation is performed on area-A, considering terrestrial APs and standard CDL and TDL channel profiles. Standard profiles A and D are used for NLoS and LoS scenarios, respectively. While the average CQI is 6 for customized CDL channel profiles, as shown in Table 8, it is found to be 9 for standard CDL channel profiles and 7 for standard TDL channel profiles. The number of SNs reporting each CQI index is illustrated in Figure 5 and Figure 6, considering the customized and standard channel profiles. The considerable difference in the CQI indices clearly shows the advantage of customizing the CDL channel profile based on the geometry of the environment to obtain more accurate results. In terms of the average network DL throughput, it is found to be 11.51 Mbps for standard CDL channel profiles and 9.04 Mbps for standard TDL channel profiles, as compared with 7.44 Mbps for customized CDL channel profiles. Table 9 summarizes the results for different channel profiles.

#### 3.4.2. Area Characterization

The area can be characterized by identifying the relationship between the CQI and SINR values. Figure 7 shows the CQI versus SINR curve for terrestrial APs considering the CDL profiles for all SNs. It can be observed that the LoS data set is clustered at higher SINR values than the NLoS data. For simplicity, each data set is modeled as a linear equation, as follows:(9)$$CQI=a_{1}\;SINR+b_{1},$$
where *a*_1_ and *b*_1_ are constants that depend on the environmental characteristics and *SINR* is (in dB). Certain exclusion rules can be imposed to increase goodness of fit. For example, the *CQI* values are 1 whenever the *SINR* is less than or equal to −25 dB for the case of *NLoS* or less than or equal to 0 dB for the case of *LoS*. On the other hand, the *CQI* values are 15 when the *SINR* values exceed 20 dB in the *NLoS* scenario, and if they exceed 45 in the case of LoS. The results of modeling the *CQI* vs. *SINR* curves will be a piecewise linear equation illustrated in (10) and (11), where ${CQI}_{NLoS}$ is for the case of *NLoS* and ${CQI}_{LoS}$ is for the case of LoS.
(10)$${CQI}_{NLoS}=\left\{\begin{array}{c} \;\;1\;\;\;\;\;\;\;\;\;\;\;\;\;\;\;\;\;\;\;\;\;\;\;\;if\;SINR\leq -25\\ 0.3975\;SINR+7.918\;\;if-25<SINR\leq 20\;\\ 15\;\;\;\;\;\;\;\;\;\;\;\;\;\;\;\;\;\;\;\;\;if\;SINR>20. \end{array}\right.$$
(11)$${CQI}_{LoS}=\left\{\begin{array}{c} 1\;\;\;\;\;\;\;\;\;\;\;\;\;\;\;\;\;\;\;\;\;\;\;\;\;\;if\;SINR\leq 0\\ 0.3822\;SINR+1.742\;\;if\;0<SINR\leq 45\;\;\;\\ 15\;\;\;\;\;\;\;\;\;\;\;\;\;\;\;\;\;\;\;\;\;\;\;\;\;\;if\;SINR>45 \end{array}\right.$$

The goodness of fit is evaluated in terms of R-squared and root mean square error (RMSE), which are found to be 0.6 and 3.1, respectively, for *NLoS* and 0.6 and 3.5 for *LoS*. It is worth noting that acceptable ranges for R-squared and RMSE depend on the application.

### 3.5. Area-B

To demonstrate that the proposed approach is comprehensive and can be generalized, another simulation is performed on area-B (Manhattan). The area of this environment is 1 km^2^, and the system is developed such that it has 42 cells located in 14 different locations and 148 SNs. The ASE is found to be 3.5 bps/Hz, with 66 SNs receiving NLoS connections, 78 are LoS, and 4 SNs are not connected. Figure 8 shows the relationship between the ASE and UAV altitude for this simulation area. Compared with area-A, a higher altitude is required to achieve the maximum ASE. This is due to the fact that area-B has higher buildings than area-A. It can be observed that increasing the UAV altitude to 145 m results in a maximum ASE of 6.776 bps/Hz, despite the fact that about 90% of the SNs in this case will be served by the UAVs. In addition, it is found that three of the remaining terrestrial APs do not serve any SN and can be relocated to serve another area.

Figure 9 shows the number of SNs reporting each CQI index, and Figure 10 shows the distribution of SNs based on the DL throughput.

Table 10 summarizes the results and shows the improvement achieved when UAVs are used in conjunction with terrestrial APs.

From the results for area-B, it can be seen that even if three terrestrial APs are removed and transmitters in 21 different cells are reduced in size, placing the APs on UAVs (higher altitude) improves the overall performance of the network.

#### 3.5.1. Customized versus Standard Channel Profiles

While the average CQI is 7 for customized CDL channel profiles in area-B, considering terrestrial APs as shown in Table 10, it is found to be 11 for standard CDL channel profiles and 9 for standard TDL channel profiles. The number of SNs reporting each CQI index is illustrated in Figure 11 and Figure 12, considering the customized and standard CDL and TDL channel profiles. Table 11 summarizes the results for different channel profiles.

#### 3.5.2. Area Characterization

Figure 13 shows the CQI versus SINR curve for terrestrial APs. Equations (12) and (13) show the constants *a*1 and *b*1 that represent the environmental characteristics, where R-squared and RMSE are found to be 0.7 and 2.5, respectively, for NLoS and 0.5 and 3.4 for *LoS*.
(12)$${CQI}_{NLoS}=\left\{\begin{array}{c} \;\;1\;\;\;\;\;\;\;\;\;\;\;\;\;\;\;\;\;\;\;\;\;\;if\;SINR\leq -25\\ 0.4097\;SINR+8.165\;if-25<SINR\leq 20\;\;\;\\ 15\;\;\;\;\;\;\;\;\;\;\;\;\;\;\;\;\;\;\;if\;SINR>20. \end{array}\right.$$
(13)$${CQI}_{LoS}=\left\{\begin{array}{c} 1\;\;\;\;\;\;\;\;\;\;\;\;\;\;\;\;\;\;\;\;\;\;\;\;\;if\;SINR\leq 0\\ 0.3622\;SINR+1.77\;if\;0<SINR\leq 45\;\;\;\\ 15\;\;\;\;\;\;\;\;\;\;\;\;\;\;\;\;\;\;\;\;\;\;\;\;if\;SINR>45 \end{array}\right.$$

### 3.6. Discussions

For both simulation areas, improvements in the RSS values are approximately 30%, and SINR values reached a range of 20 dB. In addition, the channels for all the SNs are LoS, and the ASE values are almost doubled. Although such improvements are achieved considering UAVs at an altitude of 50 m in area-A, area-B requires an altitude of 145 m for the UAVs. This is due to the geometry of area-B, which has higher buildings than area-A. In terms of average DL throughput, area-A achieved more than 40% improvement, while achievement in area-B was slightly less than 30%. A variation in DL throughput improvement is expected because it is subject to the number of SN attached to each cell and the scheduling strategy for the allocation of resource blocks among UTs. Other scheduling strategies, such as best-CQI, can be considered to improve the overall network throughput at the expense of fairness. For example, in best-CQI scheduling, resource blocks will not be allocated to UTs with low CQI indices, such as those at the cell edges. It is worth noting that the results show that the advantages of replacing some terrestrial APs with UAVs are not limited to performance improvements; they can also reduce costs because a smaller number of remaining terrestrial APs is required. After replacing 50% of the terrestrial APs with UAVs in area-A, one of the remaining terrestrial APs can be removed without affecting the performance. In contrast, UAVs in area-B can eliminate three of the remaining terrestrial APs.

## 4. Conclusions

The use of UAV-based 3D architecture in IoT networks has the potential to improve the performance of IoT applications and is paving the way for 6G new radio networks. In addition to achieving LoS and mitigating the propagation challenges of high-frequency carriers in urban areas, ease of control and relocation of UAVs can improve network scalability and flexibility. This study suggests a mechanism for modeling and evaluating UAV-assisted 3D IoT networks. The mechanism includes customizing CDL channel profiles to account for the geometry of an environment and optimizing the altitude of UAVs to achieve the highest spectral efficiency. Simulations are performed for two different urban areas, which could be the fields of deployment of various types of IoT sensor networks.

Simulation results show the effectiveness of the use of UAVs in conjunction with terrestrial APs in improving network performance in terms of various indicators, including RSS, SINR, and DL throughput. Moreover, the simulation results show that customizing CDL channel profiles can provide a considerable improvement in CQI value accuracy compared with standard channel profiles. This demonstrates the effect of the environment on the simulation results and confirms the importance of considering the geometry of the environment in the channel model. Finally, the results show that the areas can be characterized by statistical parameters. Such a characterization provides a one-to-one correspondence between the SINR and the resulting CQI values, which can simplify the evaluation of the system’s performance.

As an extension of this research, future work is planned to investigate the wireless channel and network performance of the UAV-assisted IoT network, with UAVs supporting mobility. Advanced techniques for trajectory and altitude planning and control of the UAVs will be considered to mitigate interference. In addition, practical aspects, such as channel measurements, power consumption, and wireless backhaul of UAVs, can be investigated.

## Figures and Tables

**Figure 1 sensors-24-01528-f001:**
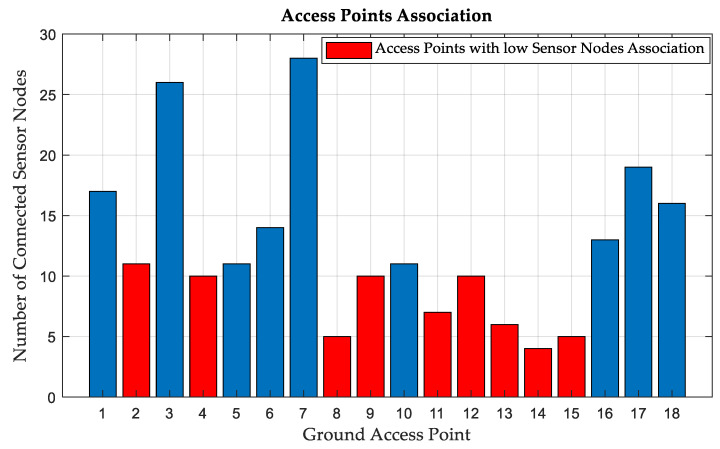
Number of sensors served by each AP site. The red bars show underperforming APs with low SNs association.

**Figure 2 sensors-24-01528-f002:**
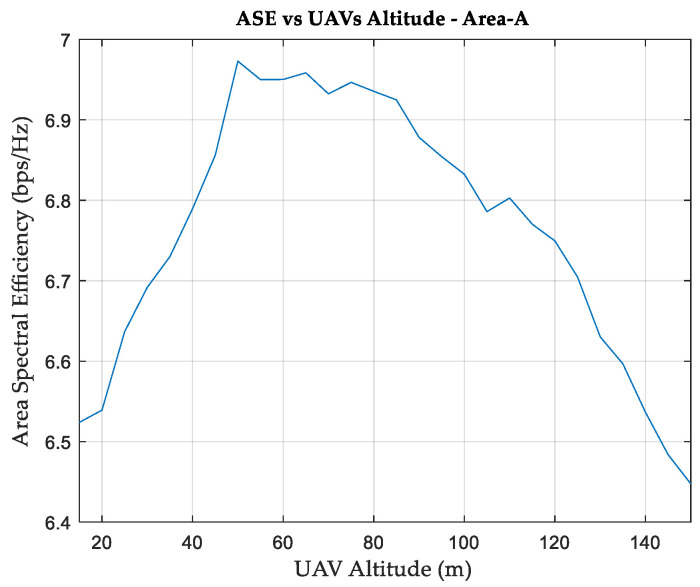
Area spectral efficiency versus UAV altitude, Area-A.

**Figure 3 sensors-24-01528-f003:**
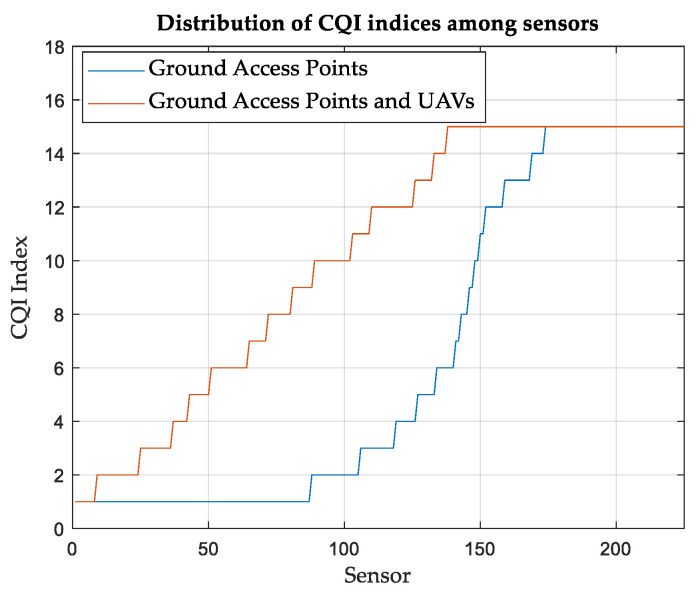
SNs distribution (ascending) based on the CQI index, Area-A.

**Figure 4 sensors-24-01528-f004:**
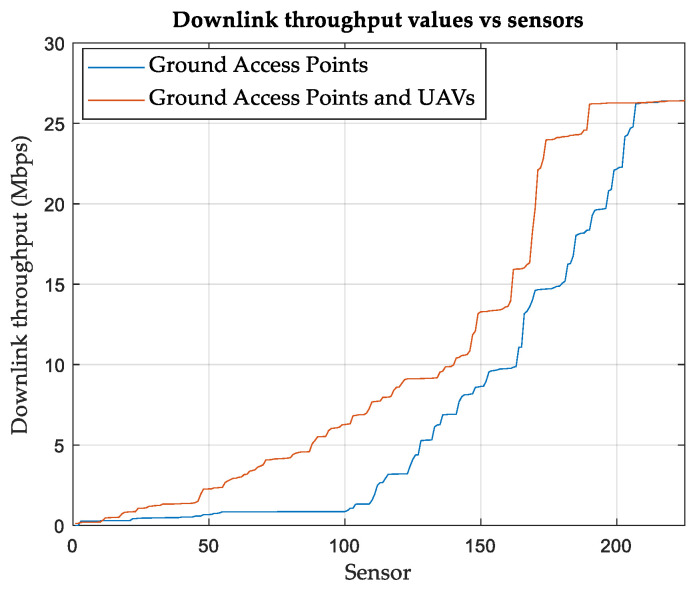
SNs distribution (ascending) based on downlink throughputs in (Mbps) using round robin scheduling, Area-A.

**Figure 5 sensors-24-01528-f005:**
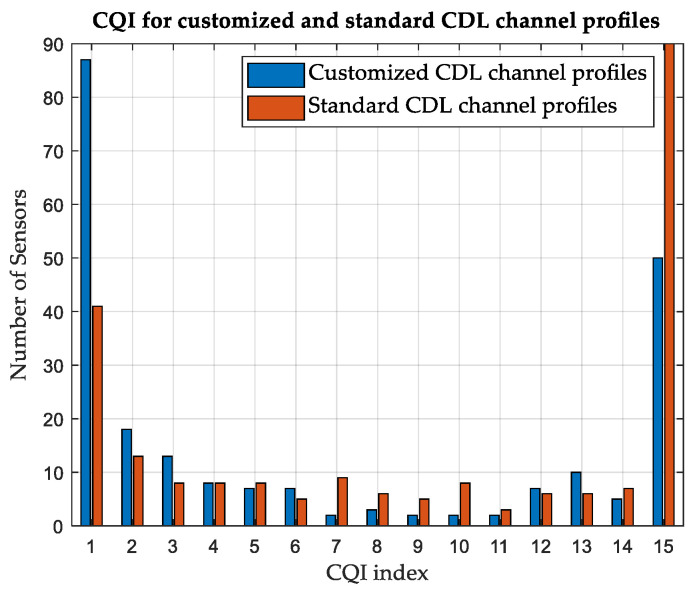
Variation in CQI between customized and standard CDL channel profiles, Area-A.

**Figure 6 sensors-24-01528-f006:**
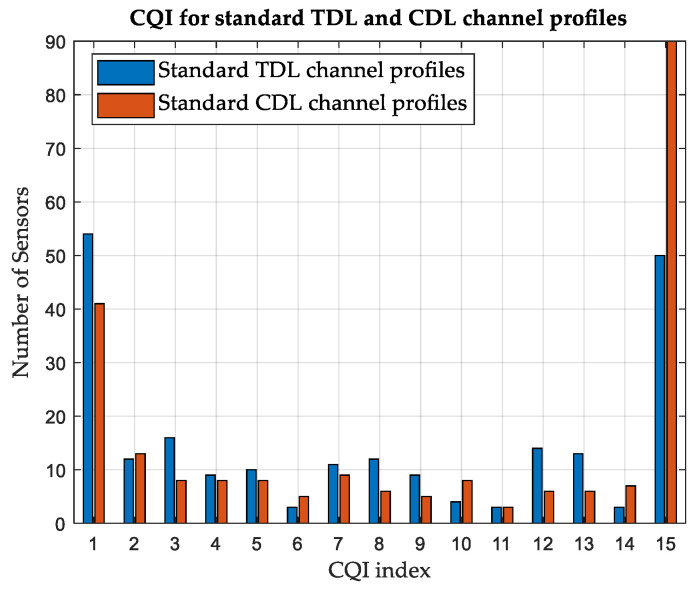
Variation in CQI between standard TDL and CDL channel profiles, Area-A.

**Figure 7 sensors-24-01528-f007:**
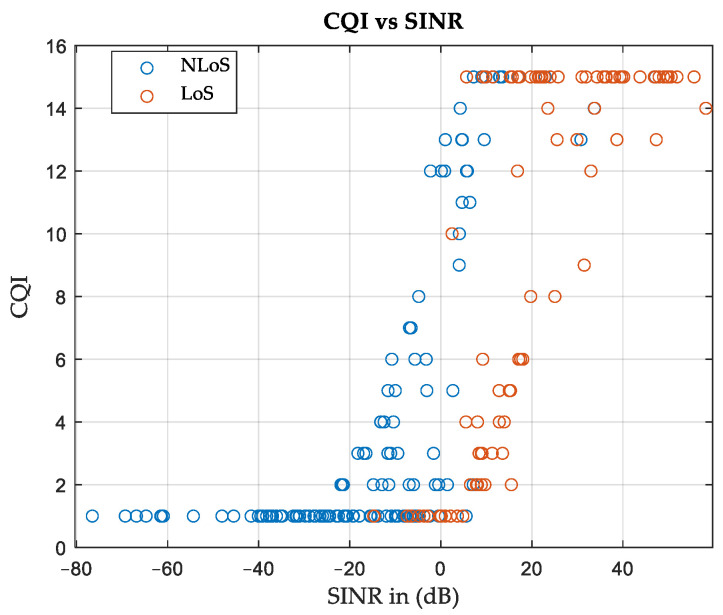
Relation between SINR values and resulting CQI, Area-A.

**Figure 8 sensors-24-01528-f008:**
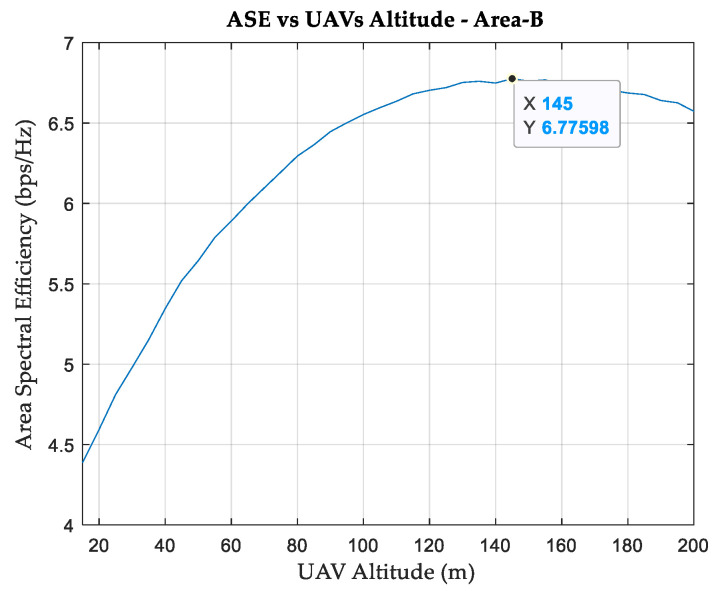
Area spectral efficiency versus UAV altitude, Area-B.

**Figure 9 sensors-24-01528-f009:**
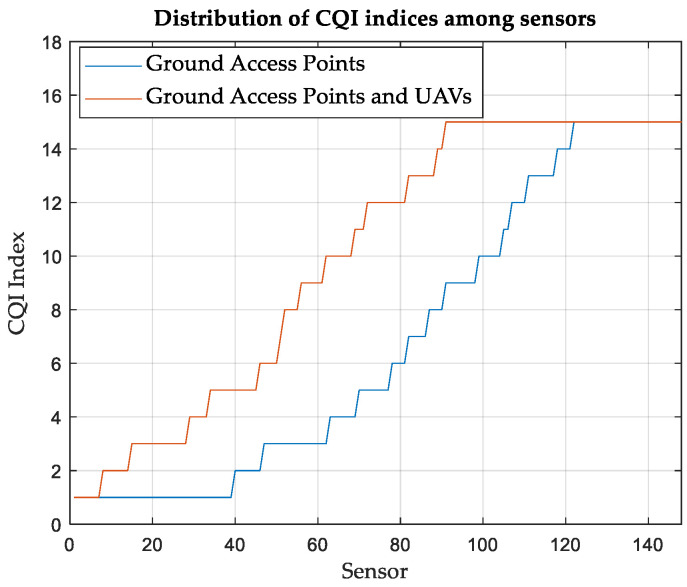
SNs distribution (ascending) based on CQI index, Area-B.

**Figure 10 sensors-24-01528-f010:**
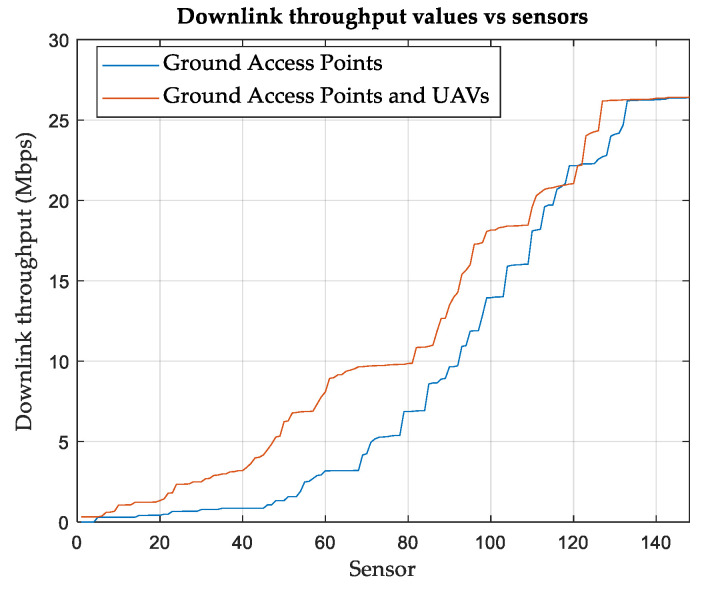
SNs distribution (ascending) based on downlink throughputs in Mbps using round robin scheduling, Area-B.

**Figure 11 sensors-24-01528-f011:**
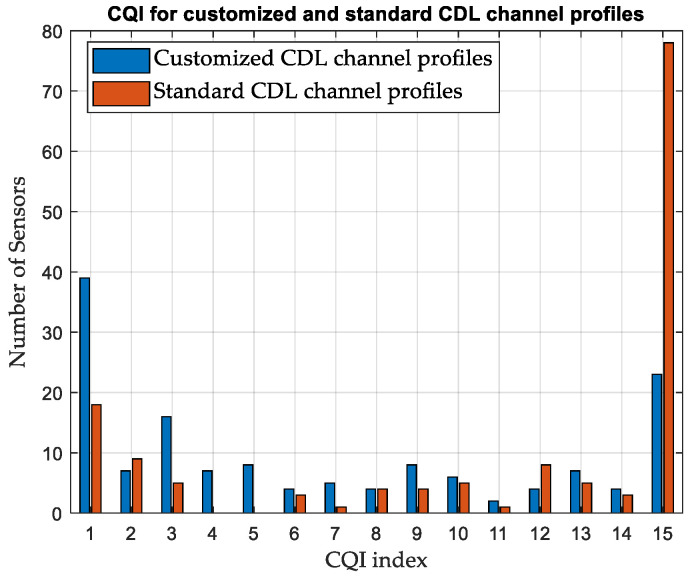
Variation in CQI between customized and standard CDL channel profiles, Area-B.

**Figure 12 sensors-24-01528-f012:**
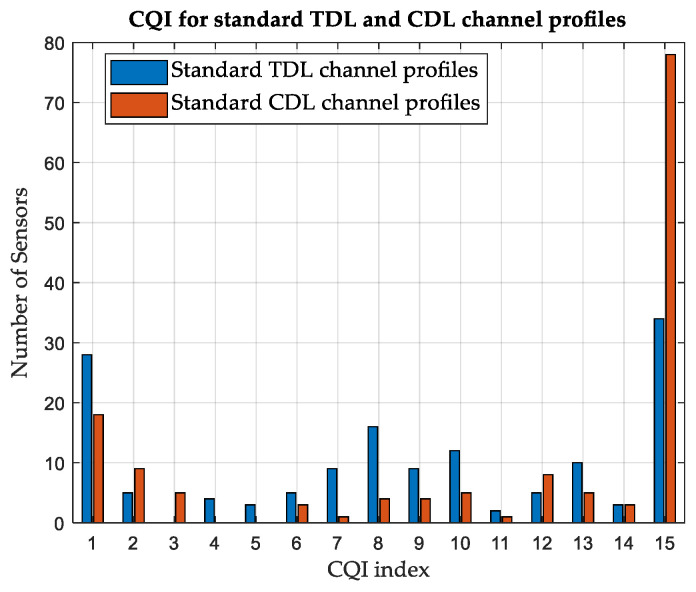
Variation in CQI between standard TDL and CDL channel profiles, Area-B.

**Figure 13 sensors-24-01528-f013:**
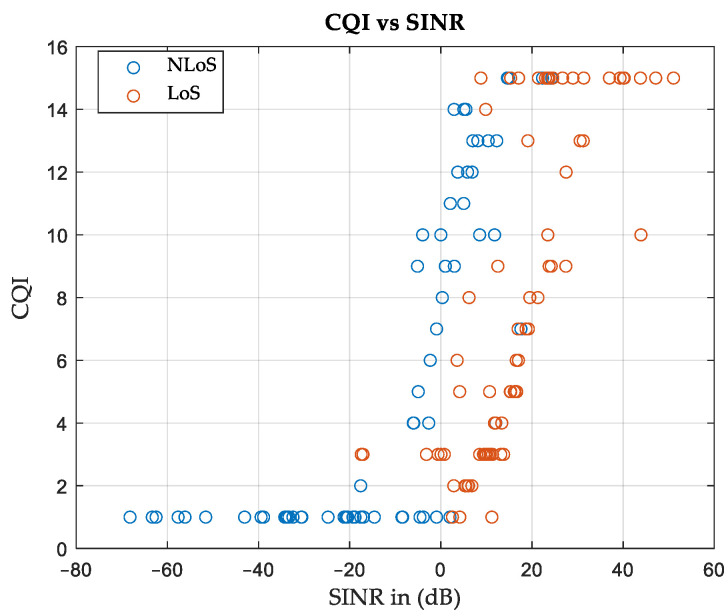
Relation between SINR values and resulting CQI, Area-B.

**Table 1 sensors-24-01528-t001:** Summary of existing frameworks in UAV-assisted communication systems.

References	Wireless Access	Objective	Mobility	Solution Approach	Channel Model
[22]	Aerial	Minimizing energy consumption of UAVs	Static	Optimize UAV positions and functional split options	Pathloss
[23]	Aerial	Maximize sum rate of UTs	Static	Optimize UT association, transmit power, and UAV placement	Pathloss
[24]	Aerial	Maximize data throughput	Mobile	Optimize travel directions of UAVs	Pathloss
[25,26,27]	Aerial	Provide cellular connectivity	Static	Study coverage probability rate	Stochastic
[29]	Aerial and terrestrial	Extend cellular connectivity	Static	Study coverage probability	Stochastic geometry-based
[30]	Aerial	Provide cellular connectivity	Static	Compares performance of different types of UAVs	Spatial (two-ray)
[31]	Aerial and terrestrial	Complement terrestrial network in breakdown scenarios	Static	Private and common message distribution	Pathloss
[13]	Aerial and terrestrial	Complement terrestrial network	Static	UAV proportion and altitude	Pathloss
[32]	Aerial and terrestrial	Complement terrestrial network and alleviate overload	Static	Optimize number of UAVs and their locations	Pathloss

**Table 2 sensors-24-01528-t002:** Channel models supporting mmWave and sub-6 GHz bands.

References	Model	Frequency Range	Type
[34]	Third Generation Partnership Project (3GPP) Geometry-based Stochastic Channel Model (GBSCM)	0.5–100 GHz	Geometry-based stochastic
[35,36]	Mobile and wireless communications Enablers for the Twenty-twenty Information Society (METIS)	2–60 GHz	Stochastic
[37]	Quasi-Deterministic Radio Channel Generator (QuaDRiGa)	0.45–100 GHz	Geometry-based stochastic
[33]	NYUSIM Channel Model	0.5–150 GHz	Spatial statistical

**Table 3 sensors-24-01528-t003:** Simulation parameters.

Notation	Description	Unit	Value
-	Cells per site	Cells	3
$f_{c}$	Carrier frequency	GHz	28
BW	Channel bandwidth	MHz	50
NRB	Number of resource blocks	Blocks	66
SCS	Subcarrier spacing	KHz	60
Fr	Number of 10 ms frames	Frames	40
Kr	Ricean K-factor	dB	13.3
ε	Buildings permittivity	F/m	3.75
σ	Buildings conductivity	S/m	0.038
$P_{tx}$	Transmitter power	dBm	44
MT × NT	Transmitter size, in case of terrestrial AP	-	8 × 8
$h_{BS}$	Transmitter height, in case of terrestrial AP	m	10
αBS	Transmitter down-tilt, in case of terrestrial AP	degrees	30
mT × nT	Transmitter size, in case of UAV	-	2 × 2
αUAV	Transmitter down-tilt, in case of UAV	degrees	75
S	Receiver sensitivity	dBm	−120
MR × NR	Receiver size	-	2 × 2
$h_{UT}$	Receiver height	m	1.5
αR	Receiver up-tilt	degrees	45
$G_{r}$	SN antenna element gain	dBi	0
NF	SN noise figure	dB	7

**Table 4 sensors-24-01528-t004:** Radio conditions of SN 1 and SN 2.

	SN-1	SN-2
Number of rays (propagation paths)	24	33
Channel type	NLoS	LoS
RSS (dBm)	−110.63	−43.22
SINR (dB)	−13.72	50.20
USE/CSE (bps/Hz)	0.060	16.677
ASE (bps/Hz)	8.368	

**Table 5 sensors-24-01528-t005:** Extracted parameters of the developed channel profile CDL-F.

Cluster	Normalized Delay	Power in [dB]	AoD in [°]	AoA in [°]	ZoD in [°]	ZoA in [°]
1	0.0000	−14.0	91.8	−79.0	94.8	85.5
2	0.0434	−19.8	121.4	−79.7	94.7	85.4
3	5.3160	−40.3	−72.5	117.2	103.2	89.8
4	5.3343	−42.5	−72.6	118.3	102.7	107.3
5	5.8522	−29.5	128.6	−46.1	92.9	86.8
6	6.3281	−34.7	67.7	146.3	92.8	86.4
7	6.3504	−36.3	67.8	155.6	93.3	105.1
8	6.3509	−36.3	67.6	142.1	93.3	106.4
9	6.5030	−1.8	50.2	−17.5	92.9	87.3
10	6.8239	−10.5	49.6	−131.2	92.8	87.4
11	6.8758	−34.6	92.1	100.5	93.3	104.6
12	7.1776	−10.4	−22.1	−20.0	92.7	87.3
13	7.4428	−43.2	86.9	91.6	92.6	87.2
14	7.4918	−18.7	−21.6	−138.5	92.7	88.6
15	7.5397	−31.6	112.4	112.8	92.6	87.1
16	8.2178	−39.3	−72.1	112.4	92.6	87.8
17	8.4773	−31.0	−21.3	−128.6	92.5	87.8
18	10.4971	−19.4	−148.5	−131.4	92.1	75.1
19	11.6704	−50.8	102.8	112.7	92.1	87.9
20	11.8856	−11.1	3.1	−14.7	92.2	88.0
21	12.2166	−21.2	2.6	−132.5	92.1	92.6
22	12.6771	−61.1	−73.7	112.6	92.1	88.7
23	12.8253	−61.1	−73.6	111.6	91.9	87.1
		Per-Cluster Parameters				
Parameter		c_ASD_ in [°]	c_ASA_ in [°]	c_ZSD_ in [°]	c_ZSA_ in [°]	XPR in [dB]
Value		10	22	3	7	8

**Table 6 sensors-24-01528-t006:** Extracted parameters of the developed channel profile CDL-G.

Cluster	Cluster PAS	Normalized Delay	Power in [dB]	AoD in [°]	AoA in [°]	ZoD in [°]	ZoA in [°]
1	Specular (LOS path)	0.0000	−0.4	130.8	−49.2	105.8	74.2
	Laplacian	0.0000	−13.7	130.8	−49.2	105.8	74.2
2	Laplacian	0.0626	−17.5	131.1	−51.0	109.4	97.4
3	Laplacian	0.0627	−17.4	130.8	−49.7	109.2	98.5
4	Laplacian	2.0752	−24.9	27.1	−32.5	100.9	79.1
5	Laplacian	2.1170	−28.3	27.1	−33.0	103.4	94.9
6	Laplacian	2.6572	−25.5	122.2	94.4	100.0	79.7
7	Laplacian	2.7499	−32.8	122.3	95.8	102.1	118.0
8	Laplacian	2.7839	−30.2	122.7	95.6	107.3	94.8
9	Laplacian	3.1780	−36.0	−175.2	−30.8	99.4	80.6
10	Laplacian	3.2118	−38.8	−175.1	−31.6	101.4	94.2
11	Laplacian	5.8166	−48.4	−8.2	−32.6	97.0	82.9
12	Laplacian	5.8358	−50.1	−8.2	−32.7	98.8	89.9
13	Laplacian	6.0239	−83.6	141.9	143.0	96.9	83.0
14	Laplacian	6.2614	−47.1	24.4	149.6	96.8	83.5
15	Laplacian	6.3216	−49.9	24.6	152.0	98.4	104.7
16	Laplacian	6.3321	−48.8	24.0	148.9	101.0	93.2
17	Laplacian	7.3826	−57.8	−173.3	150.1	96.1	83.7
18	Laplacian	7.4385	−60.4	−173.5	152.4	97.5	106.3
19	Laplacian	7.4427	−59.3	−173.0	149.7	100.0	92.0
20	Laplacian	8.1254	−40.2	117.9	−69.1	95.6	83.7
21	Laplacian	10.0236	−69.3	−10.3	148.6	95.2	85.7
22	Laplacian	10.0608	−70.2	−10.6	148.3	98.2	89.9
23	Laplacian	10.0740	−71.6	−10.1	152.1	96.2	104.0
24	Laplacian	10.9691	−51.9	117.4	106.2	94.7	84.9
25	Laplacian	11.0313	−56.9	117.5	108.1	95.9	107.5
26	Laplacian	11.4990	−71.4	−33.0	−40.2	94.5	85.3
27	Laplacian	12.3192	−79.8	−140.4	147.9	94.2	84.4
28	Laplacian	12.3425	−80.4	−140.2	147.9	97.2	88.0
29	Laplacian	12.3672	−81.8	−140.7	152.7	95.2	106.1
30	Laplacian	12.3727	−81.6	−140.4	148.5	96.2	101.7
31	Laplacian	15.6784	−91.6	−35.7	147.2	94.5	103.6
32	Laplacian	20.5731	−75.9	118.6	111.8	92.8	85.3
33	Laplacian	29.8386	−98.5	120.3	119.9	92.1	86.7
			Per-Cluster Parameters				
Parameter			c_ASD_ in [°]	c_ASA_ in [°]	c_ZSD_ in [°]	c_ZSA_ in [°]	XPR in [dB]
Value			10	22	3	7	8

**Table 7 sensors-24-01528-t007:** CQI and MCS for channel profiles F and G.

	SN-1	SN-2
CQI	2	15
Code Rate	193	948
Modulation	‘QPSK’	‘256QAM’

**Table 8 sensors-24-01528-t008:** Key performance indicators for area-A.

	Terrestrial APs	UAV-Assisted Network	Improvement
RSS mean (dBm)	−94.32	−63.87	32%
SINR mean (dB)	−0.55	20.69	3862%
LoS channels (%)	38.2	100	162%
ASE (bps/Hz)	3.417	6.973	104%
Average CQI	6	10	67%
Average Network DL throughput (Mbps)	7.44	10.7	44%

**Table 9 sensors-24-01528-t009:** Comparison between different channel profiles, Area-A.

Terrestrial APs	Customized CDL	Standard CDL	Standard TDL
Average CQI	6	9 (Percent Error = 33%)	7 (Percent Error = 14%)
Average Network DL throughput (Mbps)	7.44	11.51 (Percent Error = 55%)	9.04 (Percent Error = 22%)

**Table 10 sensors-24-01528-t010:** Key performance indicators for area-B.

	Terrestrial APs	UAV-Assisted Network	Improvement
RSS mean (dBm)	−91.21	−61.61	32%
SINR mean (dB)	2.57	20.12	683%
LoS channels (%)	78	100	28%
ASE (bps/Hz)	3.5	6.776	94%
Average CQI	7	10	43%
Average Network DL throughput (Mbps)	9.44	12	27%

**Table 11 sensors-24-01528-t011:** Comparison between different channel profiles.

Terrestrial APs	Customized CDL	Standard CDL	Standard TDL
Average CQI	7	11 (Percent Error = 57%)	9 (Percent Error = 29%)
Average Network DL throughput (Mbps)	9.44	16.4 (Percent Error = 74%)	12.52 (Percent Error = 33%)

## Data Availability

Data are contained within the article.

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
