# Peer review of "Customized Millimeter Wave Channel Model for Enhancement of Next-Generation UAV-Aided Internet of Things Networks"

_sensors, 2024, doi:10.3390/s24051528_

Round 1

Reviewer 1 Report

Comments and Suggestions for Authors

 The simulation results for the area A and B, is suggested to be compared with other existing scenario.

Reviewer 2 Report

Comments and Suggestions for Authors

The research study “A New Millimeter Wave Channel Model for Next Generation 2 UAV-Based Internet of Things Networks” is the good contribution in the domain.  However, following observation must be incorporated in revised manuscript:

---The abstract is somewhat lengthy and could be condensed for better readability and the abstract lacks specific details about the research methodology and key findings and more information should be provided about the simulation results in the abstract and the paper contains many technical terms and acronyms without clear explanations, which might make it challenging for readers who are not experts in the field. 

-- The paper should include more details about the specific real-time applications and use cases of the present study.

---The paper lacks a detailed conclusion section summarizing the key takeaways and the future directions for this research and there are numerous grammatical and typographical errors throughout the paper, which need to be corrected for clarity and the paper uses abbreviations without providing a list of abbreviations, which can be confusing for readers.

-- I suggest authors to add recent papers from fulfill gap of the study. Following references may be added in their discussion that how present work is difference in terms of techniques:

-- Channel Tracking in IRS-based UAV Communication Systems using Federated Learning”, Journal of Electrical Engineering, 74(06), pp: 521-531, 2023.

--- Future extension of the present study should be added with conclusion

Comments on the Quality of English Language

NA

Reviewer 3 Report

Comments and Suggestions for Authors

 1. It is quite unclear what is the novel contribution of the submission. It uses standard building blocks and in a classical manner, so where does the novelty come in play? The authors must expand the description of how their model differs from the existing ones (not only classical but also the novel ones!).

 2. The main problem with the submission that it is a synthetic research, which makes too strong implications. This means that the obtained data results from a simple (not in a computational meaning) simulation. And according to the obtained results, it is stated that this or that model is superior in terms of spectral efficiency (in various forms). But this tells nothing about the real-life performance of those models since the assumed 3D model cannot capture all the possible propagation effects. So, in practice, the model with worse SE can be closer to the real-life measurements, but the authors would have turned it down simply because it demonstrated poor quality. And this is a crucial problem of such research works (not only this). They give credit to the models that perform better (based on purely synthetic data), rather than the ones that are closer to the real-life measurements.

 3. The literature review is extremely shallow, and must be dramatically improved. The field of channel modeling and experimental study is a mature one, and there is a multitude of research groups working in it (especially if we are talking about new communication systems). See, for example, the research campaigns performed by Th. Rappaport scientific groups (https://ieeexplore.ieee.org/author/37066710400), and my other researchers.

 4. The authors completely ignore the fact that there are different models assumed for future generation networks, including the 3GPP 5G model and

IMT-2020 model

"Guidelines for evaluation of radio interface technologies for IMT-2020", Oct. 2017, [online] Available: https://www.itu.int/pub/R-REP-M.2412-2017.

 a set of WINNER models

P. Kyosti et al., "ST-4-027756 WINNER II D1.1.2 v1.2: WINNER II channel models", 2007, [online] Available: https://www.cept.org/files/8339/winner2%20-%20final%20report.pdf.

 QuaDRiGa Model:

S. Jaeckel, L. Raschkowski, L. Thiele, F. Burkhardt and E. Eberlein, "QuaDRiGa—Quasi determinisc radio channel generator user manual and documentation", 2017.

 NYUSIM generated model

H. Poddar, S. Ju, D. Shakya and T. S. Rappaport, "A Tutorial on NYUSIM: Sub-Terahertz and Millimeter-Wave Channel Simulator for 5G, 6G and Beyond," in IEEE Communications Surveys & Tutorials, doi: 10.1109/COMST.2023.3344671.

 and some other models, see for example:

C. -X. Wang, J. Huang, H. Wang, X. Gao, X. You and Y. Hao, "6G Wireless Channel Measurements and Models: Trends and Challenges," in IEEE Vehicular Technology Magazine, vol. 15, no. 4, pp. 22-32, Dec. 2020, doi: 10.1109/MVT.2020.3018436.

and

C. Han et al., "Terahertz Wireless Channels: A Holistic Survey on Measurement, Modeling, and Analysis," in IEEE Communications Surveys & Tutorials, vol. 24, no. 3, pp. 1670-1707, 2022, doi: 10.1109/COMST.2022.3182539.

 So, a profound comparison with (at least) those models is expected.

 5. No signal processing is present. The information about statistical inference is absent. What was the sample size? What were the obtained results' error margins?

 6. The “model” is limited to only two locations (and very specific ones), so once cannot say about wide applicability of the obtained results.

Round 2

Reviewer 2 Report

Comments and Suggestions for Authors

Now it may be accepted for further process. My all comments have addressed...

Author Response

Thank you very much for taking the time to review this manuscript and for your confirmation that all your comments have been addressed. 

Reviewer 3 Report

Comments and Suggestions for Authors

I thank the authors for the carefully performed revision. Although the response letter is very, very curt and does not give a full understanding of all the revisions that were undertaken, the revised manuscript itself now looks much more solid and informative.

There are, however, some problems with references (some of them were not identified; see, for example, section “Error! Reference source 371 not found” and downwards). Moreover, the reference section must be formatted according to the MDPI rules (up to now, it is not so). But those are minor faults that can be removed in the final versions.

Author Response

Thank you very much for taking the time to review this manuscript and for providing the feedback that the revised manuscript looks much more solid and informative. 

Kindly note that errors appear when referring to some figures and tables were due to formatting issues with the conversion from MS word to PDF. The format of the manuscript has been revised and five such errors were detected and corrected, specifically lines 371, 421, 495, 530, and 552. 

Moreover, the reference section was revised and formatted according to the MDPI rules.